# Barriers and facilitators to HIV testing among transgender people in Georgia: Qualitative study results using the COM-B Framework

Marine Gogia[1,2]*, Tamar Zurashvili[1,2], Jack DeHovitz[3], Mamuka Djibuti[2]

1 Faculty of Medicine, Ivane Javakhishvili Tbilisi State University, Tbilisi, Georgia, 2 Partnership for Research and Action for Health, Tbilisi, Georgia, 3 Department of Medicine, State University of New York, Downstate Health Sciences University, Brooklyn, New York, United States of America

* marine_gogia@yahoo.com

## Abstract

Transgender (TG) people face a disproportionately high burden of HIV globally due to stigma, discrimination, criminalization, and limited access to trans-specific healthcare. In Georgia, HIV prevalence among TG individuals is 24.1%, compared with 0.3% in the general adult population. While predictors of HIV testing have been documented, the factors influencing testing behaviors remain insufficiently understood. This study explored barriers and facilitators to HIV testing among TG in Georgia using the Capability, Opportunity, Motivation–Behavior (COM-B) model. We conducted 15 in-depth interviews (IDIs) with TG and non-binary individuals aged 18–45 years in Tbilisi between February–April 2025. Eligible participants were HIV-negative, not enrolled in pre-exposure prophylaxis (PrEP), and recruited with support from community-based organizations. Data were transcribed, coded, and thematically analyzed, with findings mapped to the COM-B framework. Participants (8 TG women, 5 non-binary, 2 TG men; median age 23) identified multiple barriers to HIV testing: limited HIV-related knowledge, low perceived risk and limited interest in testing, confidentiality fears, stigma and discrimination in healthcare settings, geographical barriers, and the influence of substance use and mental health challenges. Facilitators included awareness of free, anonymous, and confidential services; access to outreach-based and self-testing options through community organizations; HIV risk perception; community encouragement and support; social media and dating apps; recognition of testing as self-care and moral responsibility; and financial or non-financial incentives. This study demonstrates that HIV testing behaviors among TG individuals in Georgia are shaped by intersecting individual, structural, and sociocultural factors. While community-based organizations, peer support, and self-testing reduce barriers and enhance engagement, persistent challenges like stigma, low HIV literacy, geographic inequities, and substance use continue to limit uptake. Expanding decentralized, community-driven, and confidential testing services, coupled with targeted education,

**Data availability statement:** Due to the sensitive nature of interviews conducted with transgender people and other key populations, and to protect participant confidentiality, the full qualitative interview transcripts cannot be made publicly available. In addition, all interviews and transcripts are in Georgian, increasing the risk of deductive disclosure if shared publicly. De-identified, aggregated data supporting the findings of this study are available upon reasonable request and subject to ethical approval by the Institutional Review Board of the National Center for Disease Control and Public Health (NCDC), Georgia. Data access requests should be directed to the NCDC Institutional Review Board (contact: Marina Topuridze; email: irb.ncdc@gmail.com; m.topuridze@ncdc.ge).

**Funding:** The study was supported by the Fogarty International Center and the National Institute on Alcohol Abuse and Alcoholism of the National Institutes of Health (Grant D43TW011532 to MD). Funding acquisition was led by MD. The funders had no role in the study design, data collection and analysis, decision to publish, or preparation of the manuscript. The content is solely the responsibility of the authors and does not necessarily represent the official views of the National Institutes of Health.

**Competing interests:** The authors have declared that no competing interests exist.

peer-led interventions, and integration of stigma and discrimination reduction strategies, is essential to improving HIV testing coverage among TGs in Georgia.

## Introduction

TG individuals experience a disproportionate burden of HIV globally, a disparity driven by intersecting structural, social, and individual-level factors. These include pervasive stigma and discrimination, criminalization, socio-economic marginalization, violence, and limited access to gender-affirming and culturally competent healthcare services [1,2]. TG women, in particular, are among the populations most affected by HIV. A 2013 global meta-analysis estimated that TG women are 49 times more likely to be living with HIV than the general adult population [3]. More recent studies reaffirm that HIV prevalence among TG remains alarmingly high in many countries [4,5]. This elevated risk is not solely due to individual behaviors but is deeply rooted in structural barriers that restrict access to prevention, testing, and care [6].

In Eastern Europe and Central Asia (EECA), overall HIV prevalence is relatively low, averaging approximately 1.1% among adults aged 15–49 [7], yet TG face markedly higher rates due to stigma in healthcare, lack of legal gender recognition, fear of confidentiality breaches, and the absence of tailored interventions [8,9]. In Georgia, where HIV prevalence in the general adult population is also low (0.3%) [10], a 2021 quantitative study found a high HIV burden among TG—24.1% overall and 40.5% among TG women [11]. HIV testing in the past six months was reported by 76.8% of participants and predictors for testing were living alone and experiences of enacted stigma. While these associations do not in themselves establish causality, they highlight important differences within the community that may inform targeted outreach and service provision.

The present qualitative study aimed to build on the 2021 findings by exploring in greater depth the barriers and facilitators influencing HIV testing behavior among TG in Georgia. To guide this exploration, we applied the COM-B model of behavior change [12]. This approach enables thematic analysis in which identified barriers and facilitators are systematically mapped onto the COM-B domains. Such mapping can provide a clear, theory-driven interpretation of the findings and directly inform the design of multifaceted, evidence-based interventions.

### Methodology

#### Ethics statement

Ethical approval for the study was granted by the Institutional Review Board of the Georgian National Centre for Disease Control and Public Health (IRB # 2024–090), with approval dated November 27, 2024, and valid through November 27, 2025. Verbal informed consent was obtained from each participant prior to the interview, following a detailed explanation of the study objectives, topics to be discussed, and any potential risks and benefits. All digital recordings and text files are securely stored for up to two years following the completion of the study, after which they will be permanently deleted to ensure confidentiality and data security.

## Study design

This study employed a qualitative exploratory design using IDIs to investigate barriers and facilitators to HIV testing among TG individuals in Georgia. The study was guided by the COM-B model, a widely used behavior change framework that explains behavior as the result of interactions between individuals' Capability, Opportunity, and Motivation to perform a given behavior [12]. COM-B was selected because HIV testing is a complex health behavior shaped by individual knowledge and beliefs, social and structural contexts, and motivational processes, all of which are particularly salient for TG populations facing stigma and access barriers. In this study, COM-B provided a structured lens to examine how personal capacities, environmental conditions, and motivational drivers influence HIV testing decisions among TG individuals in Georgia. This framework informed the development of the semi-structured interview guide, the thematic analysis process, and the interpretation of findings.

This manuscript was prepared in accordance with the Consolidated Criteria for Reporting Qualitative Research (COREQ) checklist to enhance transparency and rigor in qualitative reporting (S1 COREQ Checklist).

## Study setting

The study was conducted in partnership with two community-based organizations serving TG and lesbian, gay, bisexual, transgender, queer, and others (LGBTQ+) populations in Georgia. One organization is a transgender-led organization dedicated to providing tailored support and advocacy for TG, focusing on health, social inclusion, and rights protection. The second is a broader LGBTQ+ organization that offers a range of services, including HIV screening, linkage to care, PrEP, community education, and capacity building. Both organizations are actively involved in human rights advocacy and work to improve access to health and social services for marginalized groups.

## Participants and recruitment

Participants were purposively selected with the assistance of social workers from community-based organizations, based on the following inclusion criteria: self-identifying as TG or non-binary, aged 18 years or older, HIV-negative status, not currently enrolled in a PrEP program, and able to provide informed consent. The rationale for including only HIV-negative participants was to specifically explore barriers and facilitators to engaging in HIV testing among those not known to be living with HIV, and to focus on behaviors relevant to HIV prevention and testing uptake. Individuals with unknown HIV status or living with HIV were excluded to reduce heterogeneity in experiences and focus on the target behavioral outcome.

We aimed to include all gender-diverse sub-groups in the study (TG women, non-binary, and TG men). The target sample size of 15 participants was determined based on anticipated data saturation and feasibility constraints within the study timeframe. Recruitment continued until no new themes emerged, indicating that thematic saturation was achieved. Two individuals initially agreed to participate but later declined; replacements were recruited.

## Data collection

The semi-structured interview guide was pilot-tested with two participants from a separate group prior to the main study to evaluate clarity, comprehensiveness, and relevance. Feedback from these pilot interviews was used to refine the guide to ensure alignment with study objectives. Data collection was conducted exclusively via Zoom between February–April 2025. Interviews lasted an average of 1 hour, and all participants received 70 Georgian Lari (GEL, approximately $25) as an incentive. Verbal informed consent was obtained from each participant prior to the interview. The consent process involved a detailed explanation of the study's objectives, topics to be discussed, potential risks and benefits, and participants' right to withdraw at any time. With participants' consent, all interviews were audio-recorded for accurate transcription and analysis (S1 Interview Guide).

### Researcher characteristics and reflexivity

Interviews were conducted by a trained qualitative researcher with prior experience working with TG communities and HIV research. The interviewer was familiar with the local context and community-based service delivery, which facilitated rapport and open discussion. Reflexive notes were taken throughout data collection and analysis to acknowledge and minimize potential researcher bias.

### Analysis

Data were analyzed using thematic analysis informed by the COM-B model to explore barriers and facilitators to HIV testing among TG participants. Interviews were audio-recorded, transcribed verbatim, and read repeatedly to ensure familiarity with the data. During the initial phase, two researchers independently read the transcripts, noted recurring patterns and early impressions, and conducted manual open coding by labeling relevant text segments reflecting participants' experiences, beliefs, and behavioral influences. Through iterative discussions, codes were compared, refined, and organized into preliminary themes, resulting in a shared codebook.

In the subsequent phase, emerging themes were organized deductively under the COM-B model's core domains: Capability, Opportunity, and Motivation. Capability included both psychological and physiological capability (e.g., HIV knowledge, health literacy, and awareness of services). Opportunity comprised physical opportunity (e.g., access, geographic factors, outreach-based testing, and HIV self-testing) and social opportunity (e.g., stigma, confidentiality concerns, and community support). Motivation was examined across reflective motivation (e.g., risk perception, self-care, and moral responsibility) and automatic motivation (e.g., fear, incentives, substance use, and trauma). This explicit mapping of codes and themes to COM-B sub-constructs ensured transparency in the application of the theoretical framework and supported methodological rigor (S1 Codebook).

To indicate the approximate frequency of responses across the 15 participants, we used the qualitative descriptors "few" (1–4 responses), "some" (5–7), "most" (8–11), "nearly all" (12–14), and "all" (15). All themes were clearly defined and named to reflect their underlying concepts. Finally, the findings were organized and presented according to the COM-B domains and shown in diagrams separately for barriers (Fig 1) and facilitators (Fig 2), adapted from Khanna et al [13]. Representative participant quotes were included to illustrate each theme, ensuring that the analysis remained grounded in the data.

### Trustworthiness

Trustworthiness was enhanced through multiple strategies, including independent coding by two researchers, iterative discussions to refine the codebook, explicit mapping of themes to the COM-B framework, and the use of verbatim participant quotes to ground interpretations in the data.

## Results

### Participant characteristics

A total of 15 transgender and non-binary individuals participated in this study. Among them, eight self-identified as TG women, five as non-binary, and two as TG men. Participants ranged in age from 18 to 45 years (median 23, mean 26).

### Barriers and facilitators to HIV testing among TG

Barriers and facilitators to HIV testing among TG participants are presented within the COM-B framework (Capability, Opportunity, and Motivation). An overview of all identified barriers and facilitators is provided in Table 1, while their distribution across COM-B domains is shown in Fig 1 (Barriers to HIV testing according to COM-B) and Fig 2 (Facilitators to HIV testing according to COM-B). The narrative Results below describe these findings within each COM-B domain.

**Global Public Health**

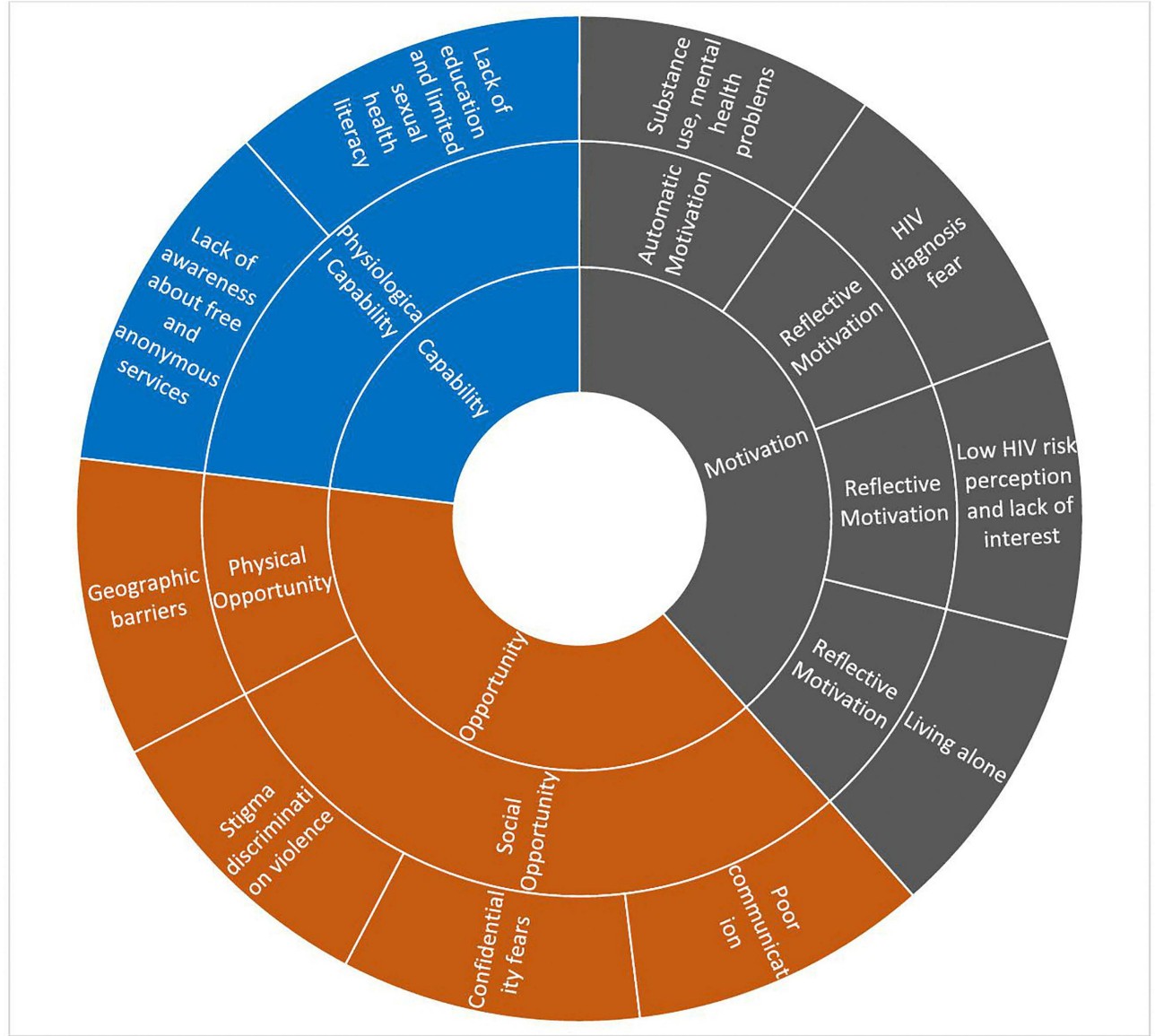

**Fig 1. Barriers to HIV testing according to COM-B.**

## Capability

Awareness/knowledge of HIV transmission risks was reported as a facilitator by nearly all (12/15) participants (quotes 1–2 in Table 1). They noted that sufficient knowledge of HIV transmission risks motivated regular testing. Awareness was shaped by personal experiences (e.g., condom failure, risky behaviors), occupational vulnerability (e.g., sex work), and exposure to public health campaigns. Community information-sharing also played a key role, with several noting that knowledge circulates quickly in small networks. However, lack of education and limited sexual health literacy remained prominent, as some participants (10/15) described that many other TG individuals lack basic knowledge about HIV, STIs, and modes of transmission (quotes 7–8), reflecting a barrier within the Psychological Capability domain of COM-B. They highlighted that some in the community are unfamiliar with HIV/AIDS, underestimate the risks of unprotected sex, or still

**Fig 2. Facilitators to HIV testing according to COM-B.**

hold misconceptions such as believing HIV can be transmitted through saliva. They linked this directly to the absence of sexual education, noting that NGOs' efforts through brochures or training are often ineffective when people have limited baseline knowledge. They noted that this limited awareness made it difficult for individuals to assess their own risk and reduced their motivation to test. Some participants (5/15) noted that many TG leave school early due to family rejection or social pressure. They explained that early school dropout, combined with broader societal neglect, restricted opportunities to learn about HIV, STIs, and the risks associated with unprotected sex.

**Table 1. COM-B Framework: Capability-related barriers and facilitators to HIV testing with supporting quotes.**

| COM-B construct | COM-B sub-construct | Barrier/ Facilitator to HIV testing | Main theme | Supporting Quote |
|---|---|---|---|---|
| Capability | Physiological Capability | Facilitator | Awareness/ knowledge of HIV transmission risks | 1."They (TGs) have information about the risks of HIV transmission and are therefore willing to get tested. It is a small community and information spreads quickly". – *Non-binary, 36 years old*<br>2."The condom broke during sex and because of that I went to get tested, I checked myself. I still do it periodically, because I think it's important". —*Trans woman, 28 years old* |
| | | | Awareness of free and anonymous services | 3."I was still in a school when I learned about these services... And others know them. CBOs that works more specifically with the trans community, are trying to make all this information and services available to TG. Previously, this information did not reach everyone, previously in 2015-2016, services were provided more to MSM people, then gradually awareness grew in trans community and among activists".—*Non-binary, 25 years old*<br>4."Yes, I know about [anonymous and confidential] testing services. It helped me that it was available, that I could get tested on the spot... Since the anonymity was maintained, I usually went and got it done... Yes, [other TG] know, everyone knows, and in my opinion, this service is the most hassle-free and accessible".—*Trans woman, 18 years old* |
| | | Barrier | Lack of awareness about free and anonymous services | 5."A barrier... In general, in 70% of cases, it's about awareness—people don't know that such services exist, and that they are free, anonymous, and confidential." — *Trans man, 23 years old*<br>6."I don't know. I would roughly say more than half know, up to 60%. It could be worse. They don't know that it's free, confidential, and they can come in anytime to get tested, and there's no charge". — *Non-binary, 33 years old* |
| | | | Lack of education and limited sexual health literacy | 7."In general, awareness is very low in Georgia, and the trans community is an especially vulnerable group. Sexually transmitted diseases, including HIV, are not taught in schools. Many TG left home very early and have not even finished school, so I think they lack knowledge about many issues, including HIV".— *Non-binary, 27 years old*<br>8."There are TG who don't know exactly how HIV is transmitted and therefore don't know the risks... They look at all of this in a way that says, "This won't happen to me and it won't be transmitted that easily".— *Trans woman, 19 years old* |
| Opportunity | Physical Opportunity | Facilitator | Good access to HIV testing sites, including outreach based HIV testing and oral self- testing | 9."I know that receiving these services is common. At certain intervals, people undergo HIV testing and not only HIV testing in Georgia… At queer events, social workers from organizations often have booths where they distribute condoms and share information… I received the self-test from an organization. I think it started during COVID, as part of a campaign by community organizations. Since we couldn't go outside, you could order a box that included condoms, lubricants, and self-tests with instructions that you could do at home and there was no need to go to center. I took my first oral test back then". — *Non-binary, 25 years old*<br>10."Oral self-testing is very easy. I've used it myself, and so has my partner. The good thing is that it helps you monitor your health and supports prevention. It's fast, and you get results immediately… Yes, I used [the saliva test]. I got it from the CBO. You can order them online, and a free taxi will deliver them. Your name isn't shown, and all personal data is protected". CBOs have their own small offices in large cities, they send testing packages".—*Trans woman, 21 years old* |
| | | | Social media and dating apps | 11."I've heard that others have found it easier to get information through social media. When they had a lot of questions and were about to take a test, posts like this helped many people, making them more confident that it's really possible to conduct such [anonymous and confidential] tests". – *Trans woman, 18 years old*<br>12."Organizations constantly post certain information online through Instagram and Facebook. Information about testing is always circulating in groups on social networks, and so on". – *Non-binary, 25 years old* |
| | | Barrier | Geographic barriers | 13."In the regions, they don't have the same resources as in Tbilisi. Referral and traveling to Tbilisi is at least a 5-hour trip for a TG person, plus transportation and additional costs". —*Non-binary, 33 years old*<br>14."Transportation is also a problem, waiting is a problem, cost is a problem...once a minibus driver threw me out because of my appearance". —*Trans woman, 21 years old* |

*(Continued)*

**Table 1.** (Continued)

| COM-B construct | COM-B sub-construct | Barrier/Facilitator to HIV testing | Main theme | Supporting Quote |
|---|---|---|---|---|
| | Social Opportunity | Facilitator | Community encourage-ment and support | 15."Trans people feel safe precisely in community organizations; there, you feel that your confidentiality will be protected. You feel like you are going to your brothers and sisters, and everything remains confidential".<br>16."Yes, I did [HIV testing]. It was much more comfortable than going to the clinic and having it done there. It was a friendlier environment, with someone encouraging you and telling you that it's not dangerous, it's very comfortable".— *Non-binary, 21 years old* |
| | | | Social stigma, dis-crimination, violence | 17."It may actually serve as encouragement and motivation, even for someone who has experienced violence. For example, a person who was raped or abused went to an NGO and got tested once, liked the service, and thought, 'Wow, testing is actually a good thing,' and now they plan to monitor their health regularly—once a month, every two or three months. So, from this perspective, it can be seen in a positive light. Of course, stigma and discrimination are bad, but they can sometimes push a person toward more frequent testing". —*Trans woman, 21 years old* |
| | | Barrier | Social stigma, dis-crimination, violence | 18."Experiences of stigma and discrimination naturally hinder people from going for HIV testing, especially if it's not at a community organization but at a medical facility. As a result, some may avoid testing. Personally, I had a similar discriminatory experience with another type of doctor. I went to one appointment, and because the environment was hostile, I didn't return for a follow-up and went to a different clinic instead… Once I avoided going to an organization because a transphobic homosexual man, known for discriminating against the trans community, was there. I wanted to participate in paid research and get tested, but he told me, 'This research is for men, for gay men who have penis…'" — *Trans man, 23 years old*<br>19."Outside of community organizations, I would feel fear, especially in places with heterosexual people. I've also heard stories of being treated harshly in medical facilities. It also depends on who you are and whether your identity is visible on the street; for TG who are easily recognized, if someone hostile notices you, going without a taxi could put you at serious risk".— *Trans woman, 18 years old* |
| | | | Fear of breach of confidentiality | 20."TG often avoid HIV testing due to fears that confidentiality will be breached and their status disclosed. Participants highlighted mistrust not only in doctors and consultants, but also within the community itself, where gossip and stigma are common. As a result, many refrain from testing, doubting that their privacy will be protected".— *Trans woman, 34 years old*<br>21."Yes, it happens often. In state clinics, nurses have shared people's HIV status with others — creating a scene and spreading panic. This happens less in NGOs but is still a concern". — *Non-binary, 21 years old* |
| | | | Poor com-munication | 22."Regarding state programs, the difficulties I've faced include not knowing who the contact persons are. Their website is poorly designed, no one ever responds to emails, and when you call, the staff are unprofessional. You have to figure everything out on your own". — *Non-binary, 27 years old*<br>23.They don't have good staff in community organizations. The staff is not motivated to develop, they look at their phones, they don't talk to us, they look at us trans people with disdain". — *Trans woman, 34 years old* |
| Motiva-tion | Reflective Motivation | Facilitator | Recognition of HIV testing as part of self-care and personal health | 24."The main motivator for testing is a health. We do it [HIV testing] to stay well and avoid serious illness". – *Trans woman, 18 years old*<br>25."I prefer to get tested and start treatment early so that I don't face health problems later and to prevent HIV from spreading in my body".– *Trans woman, 19 years old* |
| | | | Moral duty to community | 26."People who love and care for themselves tend to use testing services — it's about self-respect and protecting others, like partners, especially with frequent sex". – *Trans woman, 28 years old*<br>27."I think they want to find out whether they have the infection or not, not to spread it to others. They also conduct tests for other infections". – *Trans woman, 19 years old* |
| | | | Living alone | 28."People who have been rejected by their families tend to take better care of their health and have a stronger survival instinct". – *Trans woman, 21 years old*<br>29."When you're alone, everything is up to you. You don't want to get sick and be alone, and you want to avoid extra costs like buying medicine or going to the hospital. That's probably why they test more and pay attention to their health. Also, if I get sick, I can't go to work, and if I can't work, I might get fired". – *Trans woman, 19 years old* |

*(Continued)*

**Table 1.** (Continued)

| COM-B construct | COM-B sub-construct | Barrier/ Facilitator to HIV testing | Main theme | Supporting Quote |
|---|---|---|---|---|
| | | | High HIV risk perception | 30."People involved in sex work, especially trans women, use these services more often. Having multiple partners without knowing their status increases the risk of infection". – *Trans woman, 21 years old*<br>31."I know people who get tested quite often, especially those involved in sex work… if they learn someone they know is HIV positive, they get tested more intensively."— *Non-binary, 27 years old* |
| | | Barrier | Living alone | 32."People who live alone are often more at risk, they tend to neglect self-care. I usually don't seek help unless things get really bad. Those who live with family care more; I get tested mainly to protect my loved ones." — *Trans man, 45 years old*<br>33."If I were living alone, I probably wouldn't take care of myself as much". — *Trans man, 23 years old* |
| | | | Low perceived need and lack of interest in HIV testing | 34."Probably carelessness. They don't care about their own health or others'. They don't take it seriously. It's like they have other priorities. I think they don't know how important it is and what consequences can come from not knowing your status." — *Non-binary, 27 years old*<br>35."TG women need testing more than trans men. The trans men I know are less sexually active, often not engaging in sex without protection. Our risk is lower compared to trans women". – *Trans man, 23 years old* |
| | | | Fear of getting positive HIV test results | *36.*"I'm certain that anyone who sits down at the table to take the test feels 100% afraid, thinking, 'Oh no, what if the result comes back positive". — *Trans woman, 21 years old*<br>37."In general, everyone has that fear—that the result might come back positive. It's more about: what do I do then, how will I get treatment, what will happen to me. That's the bigger fear". — *Trans man, 23 years old* |
| | Automatic Motivation | Facilitator | Incentives (financial and non-financial) | 38."If you're paid, there's more motivation to test. Not everyone has stable income, so being compensated for caring for your health really helps". —*Trans woman, 18 years old*<br>39."Taxi fare is essential for transportation. Food and phone vouchers, hygiene items, makeup or hormones and pharmacy supplies are all things trans people need". —*Trans woman, 28 years old* |
| | | Barrier | Substance use, trauma, mental health | 40."The reason for not getting tested is probably dependence on alcohol or drugs, and the fact that many have given up on life"…there should be separate programs for them as well, to help people quit alcohol, move into treatment, and develop the motivation to see a psychologist. — *Trans woman, 19 years old*<br>41."More than half of TG are dependent on alcohol, and over half on drugs, ranging from marijuana and amphetamines to injection use. They need dedicated programs to reduce substance use and support access to psychological care." — *Trans woman, 34 years old* |

Awareness that HIV testing service in Georgia is free, anonymous, and confidential was a Psychological Capability facilitator reported by all participants, all of whom (15/15) had personal experience with HIV testing. When asked whether other TG were aware of such service, some (7/15) believed that TG community generally knows about it (quotes 3–4). Despite their own knowledge, some (8/15) participants identified lack of awareness about free and anonymous services among other TGs, particularly those in rural areas as a significant barrier (quotes 5–6).

## Opportunity

Good access to HIV services including access to outreach based HIV testing and oral self testing through trusted community-based organizations (CBOs) was reported as physical opportunity facilitator by nearly all (14/15) participants (quotes 9–10). CBOs, community events, and outreach workers were repeatedly described as crucial, providing not only testing but also information, emotional support, and safe environments that encouraged engagement. Oral self-testing was widely used, with kits and information often shared through community networks or delivered discreetly to their homes, particularly in larger cities; most participants (10/15) had personal experience with it. Although two participants expressed skepticism about the reliability of saliva-based HIV self-tests, most found them useful. Home-based and community-friendly testing was seen as easy, fast, and supportive strategies; NGO campaigns during COVID-19 further

promoted self-testing due to restrictions in travel and social gatherings. These approaches were consistently perceived as safe, stigma-free, and accessible.

Social media and dating apps were widely described as Physical Opportunity facilitators of HIV testing. Most (11/15) participants highlighted that social media platforms such as Instagram, Facebook, and community chat groups, along-side dating apps like Grindr and Tinder, serve as key channels for consistent information and targeted campaigns on HIV services by community-based organizations.These online strategies not only increased awareness but also reassured individuals about the availability and confidentiality of services. Several participants first learned about HIV testing through such channels, noting that posts, ads, or profiles of organizations encouraged them and others to get tested. For many, this digital outreach served as the first point of contact with HIV services and was seen as an accessible, low-barrier entry to care (quotes 11–12).

At the same time, geographical barriers emerged as a key Physical Opportunity barrier for most participants (10/15) (quotes 13–14), who described long travel times, high transportation costs, and safety concerns during travel, especially for visibly TG individuals, as major obstacles to HIV testing, particularly outside Tbilisi.

Experiences of social stigma, discrimination, and violence were noted as key Social Opportunity barriers by nearly all respondents (12/15). Some participants (7/15) described negative or discriminatory experiences, including humiliation, inappropriate comments, misgendering, or even physical violence, both in healthcare settings and in the broader local community outside CBO events and spaces. Such experiences, including in healthcare settings, created long-lasting avoidance, with some (7/15) reporting that after one such incident, they never returned to that facility and warned others not to go there. By contrast, CBOs were perceived as safer spaces, where fears of discrimination are reduced and where TG feel more protected and respected, saying "Going to an CBO is the only condition under which testing feels possible". Most respondents (8/13) noted that anticipated stigma plays a significant role in discouraging HIV testing. Even without direct negative experiences, the expectation of being judged or mistreated created stress, fear, and reluctance to seek testing.This was often tied to the visibility of being TG in public spaces, with those who are more visibly TG reporting greater fears of harassment or attack when traveling to testing sites. One participant directly connected increased stigma and reduced trust in both medical and community services to the challenging political climate, describing TG as "criminal-ized" and medical services as increasingly hostile (quotes 18–19). Although only one participant expressed the view that experiences of stigma and sexual violence may sometimes motivate TG to seek support from CBOs. In such cases, the safe environment and friendly staff at CBOs can reshape attitudes toward HIV testing, turning trauma into motivation for proactive health monitoring. For TG who have faced stigma or violence, trust and satisfaction with services may therefore act as facilitators for repeat testing (quote 17).

Fear of breach of confidentiality emerged as a major Social Opportunity barrier to HIV testing among TG participants. Most (10/15) participants consistently highlighted concerns that HIV status might be disclosed by medical staff, counsel-ors, or even members of the TG community, resulting in gossip, stigma, or social repercussions (quotes 20–21). Such fears were particularly acute in formal healthcare settings, including state clinics, where participants reported previous breaches of confidentiality after testing. Although CBOs were generally perceived as safer, some participants noted that the risks still exist within community organizations.

Community encouragement and support emerged as important facilitators of HIV testing. Participants described community-based organizations as safe and welcoming spaces where confidentiality is respected, stigma is absent, and interactions feel familial, fostering trust and reducing fears of discrimination (5/15 participants). Peer leaders, friends, and partners also played a key role by encouraging or accompanying individuals to test, motivating others within their net-works, and easing access through joint testing or mutual agreements (10/15 participants) (quotes 15–16).

Poor communication was also Social Opportunity barriers reported by few (2/15) participants (quotes 22–23). Issues included unresponsive staff, unclear contact points, poorly designed websites, and limited TG inclusion in outreach, which discouraged engagement with HIV testing services.

## Motivation

Recognition of HIV testing as part of self-care and personal health emerged as a Reflective Motivation facilitator among most TG individuals (9/15) to seek testing. Participants described testing as a way to protect their own health, manage risks associated with unprotected sex, and maintain overall well-being (quotes 24–25).

Moral duty to the community also emerged as a Reflective Motivation facilitator. Some participants (6/15) described testing as an act of self-respect, responsibility toward partners, and a way to protect family members. They emphasized the importance of knowing their status quickly to prevent onward transmission and to check for other infections. A few (2/15) saw themselves as motivators within their communities, encouraging peers to test and practice safer sex and increasing the sense of moral duty toward community (quotes 26–27).

Living alone emerged as a complex and ambivalent factor within the Reflective Motivation construct. For most participants (8/15), living independently was described as a facilitator of HIV testing, as it fostered greater responsibility for one's own health, heightened survival instincts (quotes 28–29). One participant linked living alone with involvement in sex work and having more frequent sexual encounters, which increases both their risk perception and the need for regular HIV testing. At the same time, few (2/15) participants mentioned living alone as a barrier, linking it to neglect of self-care and reduced motivation for testing. For these individuals, preventive behaviors such as HIV testing were often tied to protecting loved ones and accountability to family members, suggesting that the absence of social connections can reduce health-seeking behavior (quotes 32–33).

High HIV risk perception emerged as a key Reflective Motivation facilitator for HIV testing among TG participants. Nearly all respondents (14/15) emphasized that heightened risk perception either originated from involvement in sex work or from triggering events such as condom failure, condom refusal, or uncertainty about a partner's HIV status, sexual violence, or other high-risk encounters. These experiences were described as intensified feelings of vulnerability, recognition of heightened risk, and prompted immediate and more frequent HIV testing as a way to safeguard health and safety (quotes 30–31).

Low HIV risk perception and lack of interest in HIV testing were identified by most participants (10/15) as Reflective Motivation barriers (quotes 34–35). They noted that many TG individuals have low awareness of their personal risk, dislike using condoms, and deprioritize HIV testing, instead focusing on appearance or other personal matters, or assuming that HIV "does not concern them." Some (4/15) highlighted overconfidence ("it won't happen to me"), denial, or underestimation of HIV transmission risk. A few participants (3/15) described laziness or avoidance as barriers, noting that some people are indifferent or reluctant to face a potential HIV diagnosis. Few participants (2/15) emphasized perceived differences across gender identities, suggesting that TG men and non-binary people test less frequently due to lower sexual activity and, consequently, a reduced perceived need for testing.

Some of the participants (6/15) identified fear of getting positive HIV test results as a barrier within Reflective Motivation construct (quotes 36–37). In addition, one mentioned about uncertainty regarding engagement in HIV treatment following a positive test result and further health outcomes.

Our findings show that nearly all participants (14/15) agreed that incentives significantly motivate TG to undergo HIV testing. Financial incentives operate at both reflective and automatic motivation levels: reflectively, they help overcome structural barriers (transportation, poverty, daily needs), making testing more feasible and planned (quotes 62–64); automatically, they elicit emotions such as joy, urgency, or relief, prompting immediate action (quotes 65–66). Similarly, most participants (11/15) valued non-monetary incentives—taxi fares, food or phone vouchers, hygiene items, makeup, or hormone-related products.

Widespread substance use, including alcohol and drugs, was reported by nearly all participants (13/15) as a key Automatic Motivation barrier shaping HIV testing behaviors (quotes 67–70). Alcohol and drug use were reported to reduce self-control, increase engagement in high-risk sexual behaviors (e.g., chemsex, sex work, unprotected sex), and create dependency that deprioritizes routine health care. Several participants emphasized that trauma, mental health challenges,

and life stressors exacerbate this disconnection, with some noting that individuals "give up on life" or experience a sense of helplessness that diminishes motivation to seek testing. Some participants (4/15) indicated a need for integrated interventions (psychologist, psychiatrist, addiction specialist) addressing mental health needs that can contribute to increased HIV testing among them.

## Discussion

This study highlights the multifaceted barriers to HIV testing faced by TG individuals in Georgia, demonstrating how individual, structural, and sociocultural factors intersect to undermine engagement with HIV services. Key barriers identified included: (1) low HIV risk perception and lack of interest in HIV testing; (2) insufficient knowledge about HIV, and modes of transmission; (3) geographical challenges; (4) lack of awareness about free and anonymous services; (5) fear of confidentiality breaches and fear about receiving a HIV positive result; (6) social stigma, discrimination, violence (7) substance use, trauma, and mental health challenges.

Our study findings align with broader evidence indicating that low health literacy and HIV-related knowledge gaps together with lack of interest, perceived irrelevance and superficial attitude toward HIV testing can hinder HIV testing, particularly among younger TG individuals [14–17].

Evidence specifically among TG populations in the EECA region is limited. Therefore, comparisons can be drawn with other key populations and general population studies: in seven post-Soviet countries, younger and less educated women demonstrated significant gaps in HIV awareness that correlated with lower testing uptake [18]. Similarly, among sex workers in St. Petersburg, low prioritization of testing was frequently linked to limited HIV literacy and restricted access to information [19]. To address these challenges, HIV prevention strategies should integrate accessible, community-based health literacy programs tailored to TGs, focusing on TG youth and those with limited formal education.

The lack of awareness about available free and anonymous services also emerged as a significant barrier, especially among those living outside Tbilisi. Geographic isolation, limited outreach, and weak information dissemination channels contribute to a lack awareness in rural and regional areas. Similar findings have been reported in other contexts, where limited knowledge of HIV services was linked to poor dissemination of information and inadequate community-based outreach in rural settings [20]. For example, a study in Ukraine found that people in smaller towns were often unaware of HIV testing services due to insufficient public health communication and stigma surrounding testing [21]. Targeted awareness campaigns using community media, peer educators, and local networks are needed to improve visibility of HIV services in underserved regions.

Structural barriers to HIV testing include the absence of accessible services outside Tbilisi, compounded by long travel times, transportation costs, and heightened safety risks from transphobic violence.

In EECA region, HIV services are often concentrated in major cities, leaving rural and peri-urban TG populations underserved and largely invisible in health planning frameworks [22,23]. These challenges are particularly acute for economically marginalized individuals or those at elevated risk of violence when traveling. Previous studies have documented that the journey to HIV services can expose TG to stigma, harassment, or assault [24–27]. Addressing these inequities requires decentralized service hubs and active involvement of TG communities in designing secure and accessible service pathways. Mobile units, peer-led outreach, and integration into local primary care offer practical avenues to expand equitable access.

Our study revealed fear as a major emotional barrier to HIV testing, including fear of a positive result, being outed, or judgment by healthcare providers and peers. Similar patterns have been documented across the EECA region: population-level and qualitative studies report that worry about stigma, confidentiality breaches and negative treatment from providers deter people from testing or seeking confirmatory care; among sex workers, MSM and TG these anxieties frequently outweigh perceived benefits of knowing status and push people toward delaying or avoiding testing altogether [28–30]. Interventions that reduce healthcare provider stigma, guarantee confidentiality (including anonymous or

self-testing options) and address the anticipated social consequences of a positive result (peer support, linkage pathways, legal protections) are therefore likely to increase testing uptake in EECA settings [21,31,32].

Our findings confirm that stigma, discrimination, and violence represent major barriers to HIV testing among TG in Georgia, consistent with our prior quantitative research [12]. Other studies also highlight how anticipated and enacted stigma, healthcare provider discrimination, and threats to personal safety impede HIV testing in TG populations, undermine trust in services and discourage timely HIV testing and reinforced reliance on CBOs as the only safe and trusted spaces for testing [33,34]. At the same time, our data suggest a more nuanced reality: while stigma generally undermines motivation and trust, in some cases experiences of violence or discrimination heightened risk perception and prompted engagement with CBOs, where respectful environments could transform trauma into proactive health-seeking. Similar complexity in the role of stigma has been documented in other studies where community organizations served as vital access points to healthcare, turning negative experiences into opportunities for engagement by providing psychosocial support and connecting individuals to HIV prevention and treatment services [35,36]. Strengthening community-based services while addressing stigma in mainstream healthcare is therefore critical to expand Social Opportunity facilitators for HIV testing among TG populations.

Living alone was associated with increased HIV testing among TG individuals, suggesting that social isolation may heighten awareness of HIV risk and the perceived need for regular testing. However, this study offered a more nuanced view: for most participants, living alone fostered a sense of responsibility and autonomy, while for a few it acted as a barrier due to isolation and reduced motivation. Comparable dynamics have been reported in the EECA region, where social connectedness has been shown to facilitate HIV testing, and isolation or weak support networks hinder it [34,35]. Evidence from other part of world aligns with this pattern, showing that peer and partner support encourages HIV testing, while lack of social ties acts as a barrier [33,37,38]. These findings highlight that "living alone" can be both enabling and limiting, depending on social context and support networks. Programs could provide tailored support for individuals who live independently, such as peer outreach, virtual counseling, or community-based testing and self-testing to enhance motivation while mitigating isolation.

The use of alcohol and drugs, including in chemsex, contributed to induced confusion or neglect of health needs. These findings are consistent with evidence from the EECA region showing that substance use can both heighten exposure to HIV and complicate follow-up testing behaviors due to emotional distress or chaotic life circumstances. For example, a study among MSM and TG/nonbinary individuals in Kazakhstan found that polydrug use was closely linked with increased HIV risk and inconsistent testing uptake [39]. Similarly, other studies highlighted how substance use both exacerbated vulnerability and hindered engagement with HIV testing, underscoring the need for alternative approaches such as self-testing [40,41]. This dual effect emphasizes the importance of linking harm reduction services with testing opportunities in relevant settings.

This study also highlighted several key facilitators of HIV testing among TG individuals in Georgia, demonstrating how individual, structural, and sociocultural factors can enable engagement with HIV services. Consistent with existing literature, our findings indicate that engagement with HIV testing among TG were facilitated by awareness and good access to free, anonymous, and confidential HIV testing service, including outreach based HIV testing and oral self testing [42,43]. Outreach through social media and dating apps, community trust, peer influence, and interpersonal encouragement play a critical role in increasing HIV testing uptake among TG individuals in Georgia. This is consistent with WHO guidelines promoting lay provider and community-led testing strategies for key populations and highlight how social networks reduce stigma, normalize testing, and enhance engagement [44]. Evidence from the EECA region similarly demonstrates that peer-led and community-based approaches significantly improve HIV testing coverage among key populations, including TG, sex workers, and people who inject drugs, by providing trusted environments and lowering structural barriers [45,46]. In Georgia, integrated mobile outreach models operated by community organizations, resulted in a 60% increase in testing coverage among people who inject drugs [47]. Additionally, in Estonia, pilot programs of HIV rapid testing in

community settings, including needle exchange and LGBTQ+ venues, demonstrated high acceptance and effectiveness in reaching high-risk individuals [48]. These findings underscore the vital impact of community-driven, non-clinical testing strategies in overcoming barriers to build trust, foster supportive networks, and increase uptake among marginalized groups.

Consistent with existing literature, our findings indicate that at the individual level, motivation to test was strongly tied to perceived HIV risk, awareness of HIV transmission risks, personal health values, a sense of responsibility toward the community, and triggering events such as condom failure or experiences of sexual violence. Studies in Kazakhstan and Ukraine found that lack of perceived risk, competing personal priorities, and low prioritization of HIV testing hindered service uptake [35,49]. Similar dynamics have been reported in other regions, including the UK, United States, Brazil, and Western Europe, where low motivation, limited personal health values, and absence of triggering events were associated with lower HIV testing [33,39,40,50]. These findings highlight that interventions to increase HIV testing should address motivational barriers, enhance personal risk awareness to encourage proactive health-seeking behaviors.

Study findings suggest that financial and non-financial incentives, like cash, hygiene kits, transport support, food or taxi vouchers or hormone-related products, may effectively motivate economically vulnerable individuals to engage in HIV testing. Incorporating these incentives into holistic, community-driven service models may enhance uptake of HIV testing [51].

The study has some limitations. The number of participants was small, and most were living in Tbilisi/the capital, which means the findings may not fully reflect the experiences of TG living in rural or remote regions. In addition, the inclusion of only HIV-negative participants may limit the generalizability of findings to all TG individuals, particularly those with unknown or positive HIV status, and could have influenced the barriers and facilitators identified. Also, because the topic is sensitive, some participants may have given answers they thought were expected. Despite these limitations, the study provides useful insights into the barriers and motivators that affect HIV testing for TG. Future research should expand beyond capital cities and include diverse subgroups within the TG community in small cities. Applying behavior change models like COM-B in longitudinal or intervention-focused designs would also help evaluate what strategies effectively increase HIV testing uptake over time.

## Conclusion

This study highlights key barriers to HIV testing among TG individuals in Georgia. Stigma and anticipated discrimination in healthcare and public spaces were the strongest deterrents, followed by lack of education and limited sexual health literacy and low perceived risk and lack of interest in HIV testing which reduced motivation to test. Structural barriers, including service concentration in urban centers, poor awareness of free and anonymous testing, and transportation challenges, further limited access. Expanding access via decentralized services, mobile units, peer-led outreach and education, and integrating harm reduction with HIV testing can mitigate these obstacles. Implementing evidence based interventions at healthcare settings that ensure confidentiality and reduce provider stigma is critical to improving HIV testing uptake among TG populations in Georgia.

## Supporting information

**S1 Interview Guide. Semi-structured interview guide used for in-depth Interviews among transgender people.**
(DOCX)

**S1 Codebook. Codebook: COM-B Constructs, Sub-Constructs, and Main Themes.**
(DOCX)

**S1 COREQ Checklist. Consolidated Criteria for Reporting Qualitative Research.**
(DOCX)

# Acknowledgments

The authors thank local community-based organizations for assistance in recruitment of participants for this study.

# Author contributions

**Conceptualization:** Marine Gogia, Tamar Zurashvili, Mamuka Djibuti.

**Data curation:** Marine Gogia, Tamar Zurashvili.

**Formal analysis:** Marine Gogia, Tamar Zurashvili.

**Funding acquisition:** Jack DeHovitz, Mamuka Djibuti.

**Investigation:** Marine Gogia.

**Methodology:** Marine Gogia, Tamar Zurashvili, Mamuka Djibuti.

**Project administration:** Marine Gogia, Mamuka Djibuti.

**Resources:** Jack DeHovitz, Mamuka Djibuti.

**Supervision:** Mamuka Djibuti.

**Validation:** Tamar Zurashvili.

**Visualization:** Marine Gogia.

**Writing – original draft:** Marine Gogia.

**Writing – review & editing:** Marine Gogia, Tamar Zurashvili, Jack DeHovitz, Mamuka Djibuti.

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
