## [Decision Letter · Decision Letter 0]

17 Dec 2025

PGPH-D-25-02845

Barriers and Facilitators to HIV Testing among Transgender People in Georgia: Qualitative Study Results Using the COM-B Framework

Dear Dr. Gogia,

Thank you for submitting your manuscript to PLOS Global Public Health. After careful consideration, we feel that it has merit but does not fully meet PLOS Global Public Health’s publication criteria as it currently stands. Therefore, we invite you to submit a revised version of the manuscript that addresses the points raised during the review process.

We look forward to receiving your revised manuscript.

Kind regards,

Dumile Gumede, PhD (Health Promotion)

Academic Editor

Journal Requirements:

i. State what role the funders took in the study. If the funders had no role in your study, please state: “The funders had no role in study design, data collection and analysis, decision to publish, or preparation of the manuscript.”

2. Please ensure that your Ethics Statement is available in its entirety at the beginning of your Methods section, under a subheading 'Ethics Statement'.

3. We note that you have included your Figures within the body of your manuscript. Please remove the Figures from the body of your manuscript and upload them as separate Figure files.

4. Tables should not be uploaded as individual files. Please remove these files and include the Tables in your manuscript file as editable, cell-based objects. For more information about how to format tables, see our guidelines:

https://journals.plos.org/globalpublichealth/s/tables

5. We have noticed that you have uploaded Supporting Information files, but you have not included a list of legends. Please add a full list of legends for your Supporting Information files after the references list.

6. In the online submission form, you indicated that “Due to the sensitive nature of the interviews conducted with transgender and other key populations, and because all interviews and transcripts are in Georgian, full transcripts cannot be made publicly available in order to protect participant confidentiality. De-identified, aggregated data supporting the findings of this study, are available from the corresponding author upon reasonable request and subject to approval by the National Center for Disease Control and Public Health Ethics Committee.”.

3. Uploaded as supplementary information.

Additional Editor Comments (if provided):

Overall comments

This manuscript addresses an important and timely topic, barriers and facilitators to HIV testing among transgender people, which is critical for informing interventions and policies to improve HIV testing and care in this underserved population. However, there are major areas for improvement in the methodology section. Specifically, the study design is not clearly articulated, the rationale for sample size and participant selection is unclear, consent procedures are not described, trustworthiness is not addressed, and the application of the COM-B framework lacks sufficient detail. Addressing these issues would significantly enhance the rigour, transparency, and interpretability of the study.

Methodology

The methodology section lacks a clear articulation of the study design. While the theoretical framework is mentioned, the overall study design is not explicitly stated.

I appreciate that the COM-B model informed the development of the semi-structured interview guide, as indicated in the interview guide. However, the description of the framework underpinning the interview guide should be incorporated into the main manuscript rather than remaining solely in the guide. Including this information in the manuscript would help clarify the conceptual meaning of each COM-B domain and strengthen the theoretical transparency and interpretability of the study.

Study setting

Naming community-based organisations where TG individuals were recruited is not inherently unethical, but it is often unnecessary and potentially risky. Unless strong justification and explicit consent are provided, anonymising organisations is generally the more ethical approach.

Participants and Recruitment

The authors do not clearly explain how the sample size of 15 participants was determined. Please clarify the rationale for this sample size, including whether it was guided by methodological considerations such as data saturation, feasibility constraints, prior qualitative research, or established qualitative sampling principles.

The manuscript would benefit from clarification regarding the inclusion criteria related to participants’ HIV status. Specifically, please explain the rationale for including only HIV-negative individuals and excluding those with unknown HIV status, given that the study’s objective also focused on barriers to HIV testing behaviors. Additionally, this restriction should be acknowledged as a limitation, as it may have influenced the findings.

Data collection

The manuscript would benefit from clarification regarding the pilot phase. Please specify whether the pilot was conducted with the same participants included in the main study or with a separate group.

The manuscript does not specify the form of informed consent obtained from participants. Please clarify whether consent was written or verbal, and briefly describe how consent was documented.

Analysis

The authors do not clearly define the COM-B sub-constructs used in the analysis. The absence of these definitions makes it difficult to assess how the framework was applied and to evaluate the methodological rigour of the study. Please explicitly define each COM-B sub-construct and explain how they informed data collection and analysis to improve clarity and transparency.

The manuscript does not include a section addressing trustworthiness. Please add a dedicated section describing how trustworthiness was enhanced in the study.

Results

The manuscript would benefit from clarification on how participants were identified as trans men, trans women, or non-binary. Please specify whether these identities were self-reported by participants and describe the process or criteria used to categorise gender identity.

Reviewers' comments:

Reviewer's Responses to Questions

**Comments to the Author**

1. Does this manuscript meet PLOS Global Public Health’s publication criteria?

Reviewer #1: Partly

2. Has the statistical analysis been performed appropriately and rigorously?

Reviewer #1: N/A

3. Have the authors made all data underlying the findings in their manuscript fully available (please refer to the Data Availability Statement at the start of the manuscript PDF file)?

Reviewer #1: No

4. Is the manuscript presented in an intelligible fashion and written in standard English?

Reviewer #1: Yes

Reviewer #1: I appreciate the opportunity to review this manuscript presenting results from qualitative interviews with transgender people living in the country of Georgia. Given the disproportionate rates of HIV among transgender people in Georgia and other countries in the region, exploring the factors that encourage HIV testing among members of this population is important for the development and implementation of contextually appropriate HIV prevention and treatment interventions. While the manuscript provides some useful and nuances insights, areas of the methods need further clarification. Also, while the use of a theoretical framework is helpful, it also somewhat muddies the key themes and findings.

- The COM-B model needs explanation, both overall as a theoretical framework, and how it informed the design of the semi-structured interview guide and analysis. In addition, it would be helpful to know how each of the domains of COM-B are conceptualized in this context and why this was chosen as the theory of behavior change for this study. I think this could either be provided in the introduction or the methods.

- I encourage the authors to provide their final interview guide as a supplemental file.

- Further justification for the sample size is needed. Was theoretical or meaning saturation achieved?

- The analysis section is generally clear, but additional information is needed, including the make-up of the coding team, whether and how qualitative analysis software was used, and how the codebook was developed and finalized (this is hinted at, but needs to be more explicit). It would also be helpful to include the final codebook as a supplemental file.

- I appreciate the authors providing an introduction to the results and themes, but the figures made it difficult to understand the key themes vs. sub-themes, and as a result this introduction to the result was not as helpful for the reader’s understanding as it could be.

- The tables are also somewhat bulky and difficult to understand, and there are so many quotes that the voices of the participants get a little lost, I think. I think re-organizing the results by theme instead of by COM-B construct would be clearer.

- The authors may want to consider using a qualitative check-list, such as COREQ to align the manuscript with common practices in qualitative write-ups.

- Discussion does a nice job connecting the study’s findings with the literature.

- Minor but make sure to spell out abbreviations and acronyms (TG, COM-B, etc.) the first time they are used in the manuscript.

- The manuscript needs additional proofreading, as there were several places with typos and missing words that hindered readability.

**Do you want your identity to be public for this peer review?** For information about this choice, including consent withdrawal, please see our Privacy Policy

Reviewer #1: No

---

## [Decision Letter · Decision Letter 1]

17 Feb 2026

Barriers and Facilitators to HIV Testing among Transgender People in Georgia: Qualitative Study Results Using the COM-B Framework

PGPH-D-25-02845R1

Dear Dr Gogia,

We are pleased to inform you that your manuscript 'Barriers and Facilitators to HIV Testing among Transgender People in Georgia: Qualitative Study Results Using the COM-B Framework' has been provisionally accepted for publication in PLOS Global Public Health.

Best regards,

Dumile Gumede, PhD (Health Promotion)

Academic Editor

Reviewer Comments (if any, and for reference):

Reviewer's Responses to Questions

**Comments to the Author**

Reviewer #1: All comments have been addressed

publication criteria?

Reviewer #1: Yes

3. Has the statistical analysis been performed appropriately and rigorously?

Reviewer #1: N/A

4. Have the authors made all data underlying the findings in their manuscript fully available (please refer to the Data Availability Statement at the start of the manuscript PDF file)?

Reviewer #1: No

5. Is the manuscript presented in an intelligible fashion and written in standard English?

Reviewer #1: Yes

Reviewer #1: I appreciate the authors' responsiveness to the reviews. I have no further comments.

**Do you want your identity to be public for this peer review?** For information about this choice, including consent withdrawal, please see our Privacy Policy

Reviewer #1: No
